# Reasoning Models Can Be Accurately Pruned via Chain-of-Thought Reconstruction

**Ryan Lucas**[*]
LinkedIn, Sunnyvale, CA
Massachusetts Institute of Technology, Cambridge, MA

**Kayhan Behdin**
LinkedIn, Sunnyvale, CA

**Zhipeng Wang**[*]
LinkedIn, Sunnyvale, CA

**Shao Tang**
LinkedIn, Sunnyvale, CA

**Qingquan Song**
LinkedIn, Sunnyvale, CA

**Rahul Mazumder**
LinkedIn, Sunnyvale, CA
Massachusetts Institute of Technology, Cambridge, MA

## Abstract

Reasoning language models such as DeepSeek-R1 produce long chain-of-thought traces during inference time which make them costly to deploy at scale. We show that using compression techniques such as neural network pruning produces greater performance loss than in typical language modeling tasks, and in some cases can make the model *slower* since they cause the model to produce more thinking tokens but with worse performance. We show that this is partly due to the fact that standard LLM pruning methods often focus on input reconstruction, whereas reasoning is a decode-dominated task. We introduce a simple, drop-in fix: during pruning we jointly reconstruct activations from the input and the model's on-policy chain-of-thought traces. This "Reasoning-Aware Compression" (RAC) integrates seamlessly into existing pruning workflows such as SparseGPT, and boosts their performance significantly. Anonymized code can be found at: https://github.com/RyanLucas3/Reasoning-Aware-Compression

## 1 Introduction

Large Language Models (LLMs) with step-by-step reasoning abilities have become essential for solving complex, multi-step tasks in domains such as mathematics, coding, and logical reasoning (Wei et al., 2022). Reasoning models generate explicit chains-of-thought (intermediate reasoning steps) that significantly improve accuracy on challenging benchmarks, but at the cost of producing very long outputs for each query. For example, the DeepSeek-R1 model (671B parameters) (DeepSeek-AI et al., 2025) achieves strong reasoning performance but must output lengthy explanation traces, making it extremely resource-intensive to deploy at scale (Guo et al., 2025; Zhang et al., 2025a).

To reduce the cost of serving LLMs, recent years have seen a surge of interest in LLM compression techniques. Model pruning (LeCun et al., 1989; Hassibi & Stork, 1992) is a popular model compression technique where the goal is to remove redundant weights or neurons from the model, while ensuring the model quality remains high. This can lead to models with a smaller memory and compute footprint, which can be more resource efficient. Pruning has been particularly successful when applied to LLMs (Frantar & Alistarh, 2023; Meng et al., 2024; Sun et al., 2024). Given the computational requirement of reasoning LLMs, model pruning has appeared as an attractive proposition for efficient reasoning. However, the application of existing pruning methods to reasoning models can often result in significant accuracy loss (Zhang et al., 2025b), limiting the application of such compression techniques in practice where model quality is of high importance.

[*]Correspondence: ryanlucpersonal@gmail.com, zhipwang@linkedin.com

As an example, in Figure 1, we prune the DeepSeek-R1-Distill-Qwen-7B (DeepSeek-AI et al., 2025) checkpoint with SparseGPT using a single-pass C4 calibration set of 1M tokens. At sparsity levels of 30%, 50%, and 70%, we evaluate each pruned model on the MATH-500 benchmark (zero-shot, 32k max tokens). As sparsity increases, MATH-500 accuracy falls while total evaluation time grows sharply. Counter-intuitively, heavy pruning actually *slows down* inference because the model produces much longer chains of thought, rambling more yet answering less accurately. Since the goal of compression is to maintain accuracy while reducing inference time, this is obviously not desirable.

In this work, we focus on one-shot pruning of reasoning LLMs (that is, we do not conduct any retraining after pruning). One-shot pruning is well-motivated for this use-case, since while DeepSeek models are open-source (i.e., open-weights), the complete training and distillation pipeline required to retrain these models to full accuracy is not fully available (HuggingFace, 2025). Moreover, retraining is generally expensive and requires a large cluster of several GPUs. In contrast we perform all of our one-shot pruning experiments on a single H100 GPU.

We propose a new model pruning approach that better preserves the reasoning capabilities of LLMs. Our approach explicitly aligns the pruning-time reconstruction problem with the activations the model computes at inference-time (in its CoT) to generate its response. This is in contrast to existing approaches such as (Zhang et al., 2025b) that do not make use of model's CoT when pruning. Our work demonstrates that by reconstructing the CoT, reasoning LLMs can be pruned to up to 50% sparsity in one-shot accurately, maintaining up to 95% of dense model's accuracy on math and coding tasks. Additionally, our proposed method improves accuracy of pruned models on math and coding tasks by up to 17% compared to existing pruning approaches.

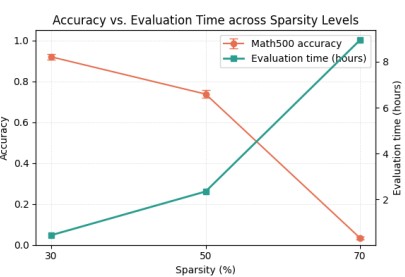

Figure 1: Pruning hurts both accuracy *and* runtime on MATH-500.

## 2 BACKGROUND

**Reasoning models.** Conventional LLMs are trained, with parameters $\theta$, to maximise the conditional likelihood $\pi_\theta(y_{0:L-1} \mid x) = \prod_{t=0}^{L-1} \pi_\theta(y_t \mid x, y_{<t})$ of an output sequence $y \in \mathcal{V}^L$ of length $L$ from the vocabulary $\mathcal{V}$ given a prompt $x$. A *reasoning model* instead produces:

$$(c_{0:T-1}, \ y_{0:L-1}), \qquad \text{with } c_{0:T-1} \in \mathcal{V}^T, \ y_{0:L-1} \in \mathcal{V}^L,$$

where $c$ is a chain-of-thought (CoT) of length $T$ and $y$ is the final answer e.g. a single numeric value, a complete proof, or a code block.[1] While a conventional LLM can be prompted to emit a CoT $c$ before producing its answer $y$, reasoning models are explicitly trained so that the generated chain and answer yield a verifiable task reward $R(x, c, y) \in [0, 1]$, for example, exact-match grading on math, unit-test passes for code, or logical consistency checks for formal proofs. This has recently become popularized by the DEEPSEEK-R1 reasoning model, which is optimized via Group-Relative Policy Optimization (GRPO) (DeepSeek-AI et al., 2025). The combined CoT and answer distribution is modeled as: $\pi_\theta(c, y \mid x) = \prod_{t=0}^{T-1} \pi_\theta(c_t \mid x, c_{<t}) \ \pi_\theta(y \mid x, c_{0:T-1})$. Given a pre-trained LLM with parameters $\theta$, GRPO fine-tunes $\theta$ by drawing $K$ full trajectories $\{(c^{(k)}, y^{(k)})\}_{k=1}^K$ for the same prompt $x$, evaluating a scalar task reward $r_k = R(x, c^{(k)}, y^{(k)})$, and computing the clipped policy-gradient loss:

$$\mathcal{L}_{\text{GRPO}}(\theta) = -\frac{1}{K} \sum_{k=1}^K \text{clip}(\rho_k(\theta), 1 - \varepsilon, 1 + \varepsilon) (r_k - \bar{r}), \qquad \bar{r} = \frac{1}{K} \sum_{k=1}^K r_k.$$

where $\rho_k(\theta) = \frac{\pi_\theta(c^{(k)}, y^{(k)} \mid x)}{\pi_{\theta_{\text{old}}}(c^{(k)}, y^{(k)} \mid x)}$ is ratio describing the probability of response $k$ in the new policy $\theta$ (being optimized) relative to the current policy $\theta_{\text{old}}$. Here, when $r_k > \bar{r}$ i.e. the $k$-th response generated higher reward than its group mean, the policy will be updated in favor of this response,

---

[1]See Wei et al. (2022) for evidence that even noisy CoT traces boost reasoning accuracy.

and vice versa. This process of generating long thought chains before outputting an answer can be viewed as training the model to perform search i.e. to try out multiple paths towards a solution before deciding on an answer. This is made explicit by generalizations of CoT such as tree-of-thoughts, which explicitly enforce a search-like procedure (Yao et al., 2023). Consequently, a longer search process (or higher number of inference tokens), has been associated with more accurate answers (OpenAI, 2024). While this makes these models more effective, it also means they incur substantial inference latency (Wei et al., 2022; Guo et al., 2025).

**LLM pruning methods.** Modern language models contain billions of parameters, so model pruning is widely used for reducing GPU memory footprint, inference latency, and energy cost while preserving most of the model's accuracy. A popular approach to model compression is model pruning via a layer-wise objective (Frantar & Alistarh, 2022). Suppose a pretrained LLM with $L$ layers is given, with layer weights $\mathbf{W}_\ell \in \mathbb{R}^{p_\ell \times d_\ell}$ for $\ell \in [L]$, and $d_\ell, p_\ell$ denote the input and output sizes of layer $\ell$, respectively. We also let $\mathbf{X}_\ell \in \mathbb{R}^{d_\ell \times N}$ denote the input activations to layer $\ell$, gathered on a *calibration dataset* of $N$ tokens. Layer-wise LLM pruning methods find compressed weights $\widehat{\mathbf{W}}_\ell$ by solving, independently for each layer:

$$\min_{\widehat{\mathbf{W}}_\ell} \left\| \mathbf{W}_\ell \mathbf{X}_\ell - \widehat{\mathbf{W}}_\ell \mathbf{X}_\ell \right\|_2^2 \text{ s.t. } \|\widehat{\mathbf{W}}_\ell\|_0 \leq S \tag{1}$$

where $\| \cdot \|_0$ denotes the number of non-zero coordinates of a matrix and $S$ is the desired number of non-zero coordinates. Numerous algorithms have been proposed for layer-wise pruning of LLMs via solving (1) (Frantar & Alistarh, 2023; Meng et al., 2024; Sun et al., 2024). For structured or semi-structured pruning, the constraint can be modified to comprise the set of block-sparse matrices or $N{:}M$ sparse matrices, and the problem can be solved with similar algorithms.

**The calibration dataset.** In Equation 1, $\mathbf{X}_\ell$ is the so-called calibration data, and for LLMs a text corpus of size $N$ tokens comprises the column dimension of $\mathbf{X}_\ell$. The calibration data is typically chosen to mimic the general distribution of natural language, e.g. the C4 dataset (Raffel et al., 2020) is a common choice. For standard LLMs, the calibration data is typically derived from a set of inputs (or prompts) $x$. Concretely, let $x_{0:N-1}$ be a batch of $N$ prompt tokens from the calibration corpus and let $E \in \mathbb{R}^{d \times |\mathcal{V}|}$ denote the embedding matrix. Define the layer-wise hidden states for each token by:

$$\mathbf{x}_t^{(0)} = E\,\mathbf{e}_{x_t}, \qquad \mathbf{x}_t^{(\ell)} = f_\ell\big(\mathbf{x}_t^{(\ell-1)}\big),\ \ell = 1, \ldots, L-1,$$

where $f_\ell$ is the $\ell^{\text{th}}$ transformer layer (including attention, MLP, residual connections, etc.), and $\mathbf{e}_{x_t}$ is the embedding for token $t$. The states are stacked column-wise to obtain the calibration activation matrix:

$$\mathbf{X}_\ell = \big[\, \mathbf{x}_0^{(\ell-1)},\ \mathbf{x}_1^{(\ell-1)},\ \ldots,\ \mathbf{x}_{N-1}^{(\ell-1)} \,\big] \in \mathbb{R}^{d_\ell \times N},$$

so the $t$-th token embedding (processed after $\ell - 1$ transformer layers) is exactly the vector that will enter layer $\ell$ when the dense model processes token $x_t$. This $\mathbf{X}_\ell$ is what standard pruning algorithms use to measure the reconstruction error of the compressed weight matrix $\widehat{\mathbf{W}}_\ell$. This setup aligns with typical LLM workloads where $|x| \gg |y|$ (long context, short reply). By contrast, in reasoning LLMs we often observe $|c| + |y| \gg |x|$ as the CoT dominates the token budget. Consequently, calibrating solely on prompts risks an inference-time distribution shift, where activations processed are primarily CoT tokens rather than input tokens.

## 3 RELATED WORK

**Benchmarking compressed reasoning LLMs.** Concurrently with our work, Zhang et al. (2025b) run an extensive evaluation of compressed DeepSeek-R1 variants. They apply SparseGPT to student models distilled from Qwen and LLaMA, but follow the default C4-style calibration pipeline. Their results are similar to what we observe: when using a generic dataset, accuracy on complex reasoning tasks drops sharply with sparsity, the CoT become repetitive or degenerate, and longer post-compression outputs correlate with lower task accuracy. Unlike their empirical study, our RAC method modifies the calibration distribution itself by injecting on-policy CoT activations, and thereby mitigates the same performance loss during pruning.

**Calibration data for post-training pruning.** Most single-pass pruning algorithms (e.g. SparseGPT, WANDA (Sun et al., 2024), ALPS (Meng et al., 2024)) rely on a small calibration set such as C4 to minimise layer-wise reconstruction loss. A recent paper reports that this choice is far from optimal. Bandari et al. (2024) compare seven pre-training and downstream datasets and finding large accuracy differences after pruning, and that across tasks C4 is rarely the optimal choice of calibration dataset. PPC-GPT (Fan et al., 2025) distils pruned student models with synthetic CoT traces, but still computes pruning scores on standard C4 activations. In contrast, we inject CoT activations directly into the pruning objective, eliminating the separate distillation stage. A related idea to ours is self-calibration (Williams et al., 2025). While the mechanism is superficially similar to RAC in that both use model-generated text as calibration data, the underlying setting is quite different. In self-calibration, the model is prompted only with a beginning-of-sequence token and allowed to generate free-form text until an end-of-sequence token or a length limit is reached. The resulting calibration distribution is therefore meant to mimic the model's pre-training text distribution and does not condition on task prompts or prune on reasoning traces.

## 4  REASONING-AWARE COMPRESSION

The core insight of RAC is to align the pruning-time reconstruction problem with the activations that the model actually computes at inference-time in order to generate the response. As such, we first recap the inference process for a reasoning LLM, and then outline our procedure for aligning the activations encountered during decoding with those used for calibration.

**Inference in an autoregressive reasoning model.** Let $x = (x_0, \ldots, x_{T_{\text{in}}-1}) \in \mathcal{V}^{T_{\text{in}}}$ be a prompt to an autoregressive reasoning model, which could be a problem of math, coding, etc., and let $\pi_\theta$ denote the dense model's conditional distribution over the vocabulary $\mathcal{V}$. At inference, a reasoning model processes and generates a completed sequence:

$$z_{0:T+L} = \big( x_0, \ldots, x_{T_{\text{in}}-1}, c_{T_{\text{in}}}, \ldots, c_T, y_{T+1}, \ldots, y_{T+L} \big),$$

which includes the prompt $x$, the chain-of-thought (CoT) tokens $c$, and the final answer tokens $y$. We partition the indices into $\mathcal{P} = \{0, \ldots, T_{\text{in}}-1\}$ being the prompt indices and $\mathcal{D} = \{T_{\text{in}}, \ldots, T+L\}$ being the decode indices. Each generated token $z_t$ (either $c_t$ or $y_t$) is drawn autoregressively,

$$z_t \sim \pi_\theta(\,\cdot \mid z_{0:t-1}\,), \qquad t \in \mathcal{D}. \tag{2}$$

To compute the generated token, for each step $t$, the model first produces an embedding for the current tokens as:

$$\mathbf{x}_t^{(0)} = E\, e_{z_t} \in \mathbb{R}^{d_1}, \quad t \in \mathcal{P} \cup \mathcal{D} \tag{3}$$

and processes them through $L$ transformer layers computing:

$$\mathbf{x}_t^{(\ell)} = f_\ell\Big(\{\mathbf{x}_\tau^{(\ell-1)}\}_{\tau \leq t}\Big), \qquad \ell = 1, \ldots, L, \quad t \in \mathcal{P} \cup \mathcal{D} \tag{4}$$

where again $f_\ell$ is the $\ell^{\text{th}}$ transformer layer. The hidden states that are required for decoding are thus:

$$\big\{ \mathbf{x}_t^{(\ell)} \; : \; \ell \in \{1, \ldots, L\},\, t \in \mathcal{P} \cup \mathcal{D} \big\}$$

That is, crucially, to generated the complete sequence, the model relies on the activations that are computed on the input, but also on activations that arise from its own self-generated tokens. Once the final hidden state $\mathbf{x}_t^{(L)}$ is computed, it is mapped to vocabulary logits via the output projection $W_{\text{out}} \in \mathbb{R}^{|\mathcal{V}| \times d_L}$ given by $\boldsymbol{y}_t = W_{\text{out}}\, \mathbf{x}_t^{(L)} \in \mathbb{R}^{|\mathcal{V}|}$. which gives the next token distribution $\pi_\theta(\,\cdot \mid z_{0:t}) = \text{softmax}(\boldsymbol{y}_t)$.

**Aligning pruning with decoding during offline calibration.** Suppose we have $M$ calibration prompts $\{x^{(m)}\}_{m=1}^M$, with corresponding prompt index sets $\mathcal{P}_m$ and decode index sets $\mathcal{D}_m$. Standard post-training compression collects activations only for $t \in \mathcal{P}_m$, i.e. only from the fixed prompt tokens in each sequence. For typical LLM applications, this is natural since in most settings $|\mathcal{P}_m| \gg |\mathcal{D}_m|$ (long context, short reply), so the majority of inference-time activations come from the prompt, and the few decode-time activations that exist are often just noisy continuations that act as proxies

for the original prompt distribution, for example, in autocomplete tasks where the completion closely mirrors the previous context. In RAC, since in reasoning tasks typically $|\mathcal{D}_m| \gg |\mathcal{P}_m|$, we modify this procedure by self-generating tokens during calibration to simulate the activations observed at decoding time. At each decode step $t$, the model's own prediction is immediately re-used as the next input:

$$z_{t+1}^{(m)} \sim \pi_\theta(\cdot \mid z_{0:t}^{(m)}), \qquad \pi_\theta(\cdot \mid z_{0:t}^{(m)}) = \text{softmax}(W_{\text{out}}\, \mathbf{x}_t^{(L,m)}), \quad t \in \mathcal{D}_m, \tag{5}$$

$$\mathbf{x}_{t+1}^{(0,m)} = E\, e_{z_{t+1}^{(m)}}, \qquad \mathbf{x}_{t+1}^{(\ell,m)} = f_\ell\Big(\{\mathbf{x}_\tau^{(\ell-1,m)}\}_{\tau \le t+1}\Big), \quad \ell = 1, \dots, L. \tag{6}$$

which gives a new set of hidden states for reconstruction $\{\mathbf{x}_{t+1}^{(\ell)}\}_{\ell=1}^L$. The resulting layer-$\ell$ input $\mathbf{x}_{t+1}^{(\ell-1,m)}$ is then appended as a new column to the decode-time activation matrix:

$$\mathbf{X}_\ell^{\text{D}} \;\leftarrow\; \big[\, \mathbf{X}_\ell^{\text{D}} \;\; \mathbf{x}_{t+1}^{(\ell-1,m)} \,\big],$$

so that after all decode steps, the activation matrix:

$$\mathbf{X}_\ell^{\text{D}} = \big[\, \mathbf{x}_t^{(\ell-1,m)} \,\big]_{\substack{t \in \mathcal{D}_m \\ m=1,\dots,M}} \in \mathbb{R}^{d_\ell \times N_{\text{D}}},$$

contains the same sequence of activations that the model will encounter during generation, where $N_{\text{D}} = \sum_{m=1}^M |\mathcal{D}_m|$. The full calibration matrix concatenates both prompt and decode activations $\mathbf{X}_\ell^{\text{RAC}} = \big[\, \mathbf{X}_\ell^{\text{P}} \;\; \mathbf{X}_\ell^{\text{D}} \,\big] \in \mathbb{R}^{d_\ell \times (N_{\text{P}}+N_{\text{D}})}$, with $N_{\text{P}} = \sum_{m=1}^M |\mathcal{P}_m|$. The RAC calibration loss for layer $\ell$ is then:

$$\|(\mathbf{W}_\ell - \widehat{\mathbf{W}}_\ell)\,\mathbf{X}_\ell^{\text{RAC}}\|_F^2 = \sum_{m=1}^M \sum_{t \in \mathcal{P}_m \cup \mathcal{D}_m} \|(\mathbf{W}_\ell - \widehat{\mathbf{W}}_\ell)\,\mathbf{x}_t^{(\ell-1,m)}\|_2^2. \tag{7}$$

The key difference between standard prompt-only calibration and RAC is that RAC's calibration set covers $\big\{ \mathbf{x}_t^{(\ell-1,m)} : \ell = 1, \dots, L,\ t \in \mathcal{P}_m \cup \mathcal{D}_m,\ m = 1, \dots, M \big\}$, i.e. all activations used during the full on-policy rollout. This set is generated by the same autoregressive mechanism (2)–(4) used at inference, so the empirical distribution over these activations during calibration matches the inference-time distribution. Since the activations are collected under the dense model's own policy, the procedure is analogous to on-policy distillation in reinforcement learning (Agarwal et al., 2024).

---

**Algorithm 1:** Reasoning-Aware Compression (RAC)

**Input:** $\{x^{(m)}\}_{m=1}^M$: calibration prompts; $f_\theta$: dense LM with $L$ layers; $T_{\max}$: max decode length; $S$: target sparsity

**Output:** $\widehat{f}$: compressed model

1 **Phase I: Activation collection**
2 **for** $m = 1, \dots, M$ **do**
    // Prompt phase
3     **for** $t \in \mathcal{P}_m$ **do**
4         compute $\mathbf{x}_t^{(0,m)}, \mathbf{x}_t^{(\ell,m)}$ via (3)–(4); append $\mathbf{x}_t^{(\ell-1,m)}$ to $\mathbf{X}_\ell^{\text{P}}$
    // Decode phase
5     **for** $t \in \mathcal{D}_m$ **do**
6         sample $z_t^{(m)} \sim \pi_\theta(\cdot \mid z_{0:t-1}^{(m)})$ via (2); compute $\mathbf{x}_t^{(0,m)}, \mathbf{x}_t^{(\ell,m)}$ via (3)–(4); append $\mathbf{x}_t^{(\ell-1,m)}$ to $\mathbf{X}_\ell^{\text{D}}$

7 **Phase II: Layer-wise compression**
8 **for** $\ell = 1, \dots, L$ **do**
9     $\mathbf{X}_\ell^{\text{RAC}} \leftarrow [\mathbf{X}_\ell^{\text{P}}, \mathbf{X}_\ell^{\text{D}}]$
10     $\widehat{\mathbf{W}}_\ell \leftarrow \text{PRUNE}(\mathbf{W}_\ell, \mathbf{X}_\ell^{\text{RAC}}, S)$         // e.g., SparseGPT, WANDA
11 **return** $\widehat{f} = \{\widehat{\mathbf{W}}_1, \dots, \widehat{\mathbf{W}}_L\}$

---

## 5 EXPERIMENTS

### 5.1 ONE-SHOT PRUNING

**Experimental Setup.** To test the effectiveness of RAC, we perform one-shot pruning on two families of open-source reasoning models: (i) DeepSeek-R1 distilled Qwen variants (1.5B, 7B, 14B, 32B) released by DeepSeek-AI et al. (2025), and (ii) the Qwen3 reasoning models (1.7B, 8B, 14B). Each model is pruned in one-shot with SparseGPT at layer-wise unstructured sparsity of 20%, 30%, 40% and 50% using 1M calibration tokens from: (i) the standard English–web C4 corpus, (ii) OPEN-R1-MATH-220K (HuggingFace, 2025) or CODEFORCES (Shanghaoran Quan, 2025) problem statements without answers or reasoning traces ("prompt only"), and (iii) those prompts augmented with up to $T_{max} = 8192$ on–policy chain–of–thought tokens collected from each corresponding dense model, as described in Algorithm 1 ("RAC").

For mathematical reasoning we use MATH500 and report **acc@1:1**: the percentage of problems for which the model's single most-confident prediction exactly matches the ground-truth answer (Top-1 accuracy). For **code generation** we use the CodeGen evaluation harness and report **acc@1:16**, i.e. the percentage of cases in which the correct solution appears anywhere within the model's top-16 predictions, irrespective of rank (Top-16 accuracy). All evaluations are zero-shot with no additional few-shot examples, and a 32k output token budget. This follows the evaluation pipeline used by DeepSeek-AI et al. (2025) and the open-source replication from HuggingFace (2025).

Table 1: DeepSeek-R1 Qwen MATH500 **acc@1:1** under one-shot pruning. Accuracy with standard error (SE) on the left, total evaluation time in minutes on the right. Best accuracy or fastest runtime in **green**.

| Model | Sparsity | Accuracy ± SE | | | Runtime (min) | | |
|---|---|---|---|---|---|---|---|
| | | C4 | Prompt-Only | RAC | C4 | Prompt | RAC |
| 1.5B | *Dense* | **0.832** | **0.832** | **0.832** | **22.6** | **22.6** | **22.6** |
| | 20% | 0.822 (0.017) | **0.840 (0.016)** | 0.832 (0.017) | 25.6 | 24.4 | **22.6** |
| | 30% | 0.762 (0.019) | 0.788 (0.018) | **0.822 (0.017)** | 31.6 | 49.9 | **25.8** |
| | 40% | 0.658 (0.021) | 0.728 (0.020) | **0.774 (0.019)** | 57.7 | 65.9 | **32.1** |
| | 50% | 0.356 (0.021) | 0.496 (0.022) | **0.664 (0.021)** | 156.5 | 154.8 | **56.7** |
| 7B | *Dense* | **0.936** | **0.936** | **0.936** | **23.3** | **23.3** | **23.3** |
| | 20% | 0.902 (0.013) | 0.928 (0.012) | **0.934 (0.011)** | 23.5 | 23.5 | **21.7** |
| | 30% | 0.904 (0.013) | 0.922 (0.012) | **0.934 (0.011)** | 27.4 | 26.8 | **25.2** |
| | 40% | 0.890 (0.014) | 0.898 (0.014) | **0.912 (0.013)** | 38.3 | 37.4 | **29.1** |
| | 50% | 0.744 (0.020) | 0.812 (0.017) | **0.900 (0.013)** | 135.0 | 115.6 | **35.3** |
| 14B | *Dense* | **0.941** | **0.941** | **0.941** | **50.3** | **50.3** | **50.3** |
| | 20% | 0.952 (0.010) | 0.954 (0.009) | **0.962 (0.009)** | 58.4 | 56.5 | **54.7** |
| | 30% | 0.936 (0.011) | 0.930 (0.011) | **0.936 (0.011)** | 67.2 | 64.9 | **58.7** |
| | 40% | 0.910 (0.013) | 0.928 (0.012) | **0.942 (0.010)** | 97.9 | 73.1 | **66.3** |
| | 50% | 0.878 (0.015) | 0.880 (0.015) | **0.910 (0.013)** | 171.4 | 124.1 | **84.9** |
| 32B | *Dense* | **0.942 (0.011)** | **0.942 (0.011)** | **0.942 (0.011)** | **64.3** | **64.3** | **64.3** |
| | 20% | **0.950 (0.010)** | 0.942 (0.011) | 0.946 (0.010) | 61.4 | 57.6 | **58.4** |
| | 30% | 0.940 (0.011) | 0.940 (0.011) | **0.954 (0.009)** | 79.0 | 67.6 | **66.8** |
| | 40% | 0.918 (0.012) | 0.934 (0.011) | **0.940 (0.011)** | 100.2 | 89.7 | **70.8** |
| | 50% | – | **0.924 (0.012)** | **0.924 (0.012)** | – | 174.0 | **100.6** |

Table 2: Qwen3 MATH500 accuracy under one-shot pruning with SparseGPT. Accuracy with standard error (SE) on the left, total evaluation time in minutes on the right. Best accuracy or fastest runtime in **green**.

| Model | Sparsity | Accuracy ± SE | | | Runtime (min) | | |
|---|---|---|---|---|---|---|---|
| | | C4 | Prompt-Only | RAC | C4 | Prompt | RAC |
| 1.7B | *Dense* | **0.906** | **0.906** | **0.906** | **18.5** | **18.5** | **18.5** |
| | 30% | 0.822 (0.017) | 0.874 (0.015) | **0.880 (0.015)** | 58.6 | 33.3 | **24.1** |
| | 40% | 0.346 (0.021) | 0.764 (0.019) | **0.858 (0.016)** | 262.6 | 87.2 | **33.1** |
| | 50% | – | 0.470 (0.022) | **0.648 (0.021)** | – | 274.5 | **101.0** |
| 8B | *Dense* | **0.962** | **0.962** | **0.962** | **41.3** | **41.3** | **41.3** |
| | 30% | 0.948 (0.010) | 0.958 (0.009) | **0.972 (0.007)** | **29.6** | 35.6 | **29.6** |
| | 40% | 0.906 (0.013) | 0.944 (0.010) | **0.968 (0.008)** | 57.5 | 38.0 | **29.4** |
| | 50% | 0.564 (0.022) | 0.470 (0.022) | **0.862 (0.015)** | 258.8 | 274.5 | **17.1** |
| 14B | *Dense* | **0.972** | **0.972** | **0.972** | **41.2** | **41.2** | **41.2** |
| | 30% | 0.950 (0.010) | 0.964 (0.008) | **0.970 (0.008)** | 26.7 | **22.6** | 23.6 |
| | 40% | 0.958 (0.009) | 0.962 (0.009) | **0.970 (0.008)** | **26.1** | 33.8 | 31.3 |
| | 50% | 0.830 (0.017) | 0.932 (0.011) | **0.962 (0.009)** | 64.4 | 43.0 | **31.6** |

Table 3: DeepSeek-R1 Llama MATH500 **acc@1:1** under one-shot pruning. Accuracy with standard error (SE) on the left, total evaluation time in minutes on the right. Best accuracy or fastest runtime in **green**.

| Model | Sparsity | Accuracy ± SE | | | Runtime (min) | | |
|---|---|---|---|---|---|---|---|
| | | C4 | Prompt-Only | RAC | C4 | Prompt-Only | RAC |
| 8B | *Dense* | **0.866 (0.015)** | **0.866 (0.015)** | **0.866 (0.015)** | **8.3** | **8.3** | **8.3** |
| | 30% | 0.860 (0.016) | 0.850 (0.016) | **0.884 (0.014)** | 9.4 | 9.6 | **8.9** |
| | 40% | 0.794 (0.018) | 0.804 (0.018) | **0.834 (0.017)** | 14.8 | 14.6 | **10.3** |
| | 50% | 0.508 (0.022) | 0.654 (0.021) | **0.714 (0.020)** | 45.5 | 30.8 | **19.3** |
| 70B | *Dense* | **0.954 (0.009)** | **0.954 (0.009)** | **0.954 (0.009)** | **13.0** | **13.0** | **13.0** |
| | 30% | 0.936 (0.011) | **0.946 (0.010)** | **0.946 (0.010)** | 16.4 | 20.2 | **13.4** |
| | 40% | 0.910 (0.013) | **0.936 (0.011)** | 0.932 (0.011) | 20.7 | 20.1 | **15.1** |
| | 50% | 0.818 (0.017) | 0.892 (0.014) | **0.904 (0.013)** | 40.9 | 39.8 | **26.6** |

Table 4: Qwen3 AIME-25 accuracy under one-shot pruning with SparseGPT. Accuracy with standard error (SE) on the left, total evaluation time in minutes on the right. Best accuracy or fastest runtime in **green**.

| Model | Sparsity | Accuracy ± SE | | | Runtime (min) | | |
|---|---|---|---|---|---|---|---|
| | | C4 | Prompt-Only | RAC | C4 | Prompt | RAC |
| 1.7B | *Dense* | **0.333** | **0.333** | **0.333** | **9.9** | **9.9** | **9.9** |
| | 30% | 0.267 (0.082) | **0.300 (0.085)** | 0.267 (0.082) | 14.5 | 12.4 | **11.4** |
| | 40% | 0.000 (0.000) | 0.133 (0.063) | **0.267 (0.082)** | 27.2 | 17.5 | **10.2** |
| | 50% | 0.000 (0.000) | 0.000 (0.000) | **0.200 (0.074)** | 26.9 | 28.3 | **13.3** |
| 8B | *Dense* | **0.633** | **0.633** | **0.633** | **12.4** | **12.4** | **12.4** |
| | 30% | 0.600 (0.091) | 0.633 (0.090) | **0.667 (0.088)** | 15.3 | **15.0** | 15.9 |
| | 40% | 0.533 (0.093) | **0.633 (0.090)** | **0.633 (0.090)** | 21.8 | **14.9** | 15.2 |
| | 50% | 0.033 (0.033) | 0.300 (0.085) | **0.500 (0.093)** | 37.4 | 25.7 | **16.5** |
| 14B | *Dense* | **0.667** | **0.667** | **0.667** | **13.9** | **13.9** | **13.9** |
| | 30% | **0.700 (0.085)** | 0.667 (0.088) | 0.667 (0.088) | **14.2** | 14.4 | 14.3 |
| | 40% | 0.500 (0.093) | 0.633 (0.090) | **0.667 (0.088)** | 16.6 | **14.4** | 17.2 |
| | 50% | 0.300 (0.085) | 0.433 (0.092) | **0.600 (0.091)** | 23.1 | 20.1 | **16.5** |

Table 5: DeepSeek-R1 Qwen LiveCodeBench **codegen_pass@1:16** under one-shot pruning. Accuracy with standard error (SE) on the left, total evaluation time in minutes on the right. Best accuracy or fastest runtime in **green**.

| Model | Sparsity | Accuracy ± SE | | | Runtime (min) | | |
|---|---|---|---|---|---|---|---|
| | | C4 | Prompt-Only | RAC | C4 | Prompt | RAC |
| 1.5B | *Dense* | **0.161** | **0.161** | **0.161** | – | – | – |
| | 20% | 0.148 (0.018) | **0.156 (0.019)** | 0.155 (0.018) | 305.7 | 511.9 | **299.4** |
| | 30% | 0.127 (0.017) | 0.138 (0.018) | **0.150 (0.018)** | 724.4 | 685.3 | **274.5** |
| | 40% | 0.066 (0.011) | 0.086 (0.013) | **0.129 (0.017)** | 466.8 | 394.5 | **277.3** |
| | 50% | 0.004 (0.002) | 0.024 (0.006) | **0.093 (0.014)** | 464.2 | 440.3 | **330.5** |
| 7B | *Dense* | **0.374** | **0.374** | **0.374** | – | – | – |
| | 20% | 0.362 (0.025) | **0.367 (0.025)** | 0.364 (0.025) | 591.4 | 356.6 | **318.7** |
| | 30% | 0.335 (0.025) | 0.341 (0.025) | **0.361 (0.025)** | 348.6 | 343.4 | **325.2** |
| | 40% | 0.273 (0.023) | 0.300 (0.024) | **0.333 (0.024)** | 485.7 | 847.8 | **344.7** |
| | 50% | 0.099 (0.014) | 0.228 (0.021) | **0.283 (0.023)** | 692.9 | 1174.8 | **381.2** |
| 14B | *Dense* | **0.513** | **0.513** | **0.513** | – | – | – |
| | 20% | 0.508 (0.027) | 0.508 (0.027) | 0.508 (0.027) | **1202.2** | 1246.6 | 1254.6 |
| | 30% | 0.491 (0.027) | **0.496 (0.027)** | **0.496 (0.027)** | 1292.1 | 1272.8 | **759.0** |
| | 40% | 0.447 (0.026) | 0.471 (0.027) | **0.480 (0.027)** | 1665.4 | 1441.2 | **1416.1** |
| | 50% | 0.319 (0.024) | 0.385 (0.026) | **0.424 (0.026)** | 1918.8 | 2472.9 | **1814.8** |

**Discussion of pruning results.** Table 1, Table 2 and Table 3 report MATH500 acc@1:1 and runtime across the DeepSeek-R1 (Qwen and Llama families) and Qwen3 reasoning families under one-shot pruning with SparseGPT. Across all architectures, RAC consistently outperforms generic C4 calibration and typically improves over prompt-only calibration, especially at higher sparsity levels. On DeepSeek-R1-1.5B at 50% sparsity, RAC attains 0.664 accuracy versus 0.496 for prompt-only and 0.356 for C4. A similar pattern holds for the 7B model, where at 50% sparsity RAC maintains 0.900 accuracy compared with 0.812 (prompt-only) and 0.744 (C4), and again substantially reduces the runtime (from 135.0 and 115.6 minutes to 35.3 minutes). For the larger 14B and 32B DeepSeek-

R1 variants, RAC preserves dense-level accuracy much better at high sparsity, while avoiding the extreme runtime blow-ups observed under other forms of calibration.

The Qwen3 results in Table 2 show that this behavior generalizes to an independent reasoning architecture. For Qwen3-1.7B and 8B, RAC consistently achieves the highest accuracy at 30–50% sparsity, often by a wide margin. For instance, at 50% sparsity on Qwen3-8B, RAC achieves $0.862$ accuracy compared to $0.564$ (C4) and $0.470$ (prompt-only), while reducing evaluation time from over 250 minutes to 17.1 minutes. Even on the strongest Qwen3-14B model, RAC closely tracks the dense baseline at 40–50% sparsity (e.g., $0.962$ vs. $0.972$ dense at 50% sparsity), whereas C4 calibration degrades much more sharply. At more moderate sparsity levels (20% on DeepSeek-R1, 30% on Qwen3), all methods remain close to the dense baseline, indicating that the benefits of reasoning-aware calibration become most pronounced as compression becomes more aggressive.

Table 4 extends this comparison to the AIME-25 benchmark, which consists of harder competition-style math problems. Under C4 calibration, accuracy often collapses at 40–50% sparsity (e.g., Qwen3-1.7B at 40% sparsity drops to $0.000$), while prompt-only traces partially mitigate this degradation but still underperform. RAC, by contrast, maintains a large fraction of dense accuracy in the high-sparsity regime. These results suggest that aligning calibration with on-policy reasoning traces is particularly important on challenging benchmarks.

Finally, Table 5 shows the LIVECODEBENCH `codegen_pass@1:16` results for DeepSeek-R1. The qualitative pattern mirrors the math experiments: code generation is highly sensitive to pruning, C4 calibration suffers severe accuracy and runtime degradation at high sparsity, prompt-only calibration helps but is insufficient, and RAC reduces the loss from pruning the most. Across Math500, AIME, and LiveCodeBench, domain-relevant calibration (prompt-only) consistently outperforms generic C4, but RAC provides the most robust pruned models, especially at high sparsity. Larger models (e.g., 14B and above) are intrinsically more robust to pruning, but still benefit meaningfully from RAC in both accuracy and runtime.

**RAC mitigates errors during decoding.** The purpose of this experiment is to test whether RAC reduces reconstruction error on unseen reasoning problems, not just those encountered during calibration. To that end, we evaluate pruned DeepSeek-R1-Distill-7B models on the MATH500 test set, which contains problems the model did not encounter during calibration. For each problem we construct a fixed evaluation sequence: the concatenation of (1) the original prompt $x_{0:T_{\text{in}}-1}$ and (2) the dense model's own greedy rollout $c_{T_{\text{in}}:T}$. At each step $t$ we feed this shared prefix $z_{0:t}$ to the dense model and to a pruned model, take the last-layer, last-token hidden state, and compute the tokenwise reconstruction error $e_t^{(\text{meth})} = \left\| \mathbf{x}_{\text{dense},t}^{(L)} - \mathbf{x}_{\text{meth},t}^{(L)} \right\|_2, t = 1, \ldots, T$ where "meth" refers to pruning with either prompt-only calibration or RAC. The heatmap then plots the ratio $r_t = e_t^{(\text{Prompt})} / e_t^{(\text{RAC})}$, row-wise over problems (columns are token indices). The dashed black line marks the prompt/decoding boundary $t = T_{\text{in}}$. Grey cells ($r_t \approx 1$) indicate both methods incur similar error. Blue ($r_t > 1$) indicates RAC achieves smaller error. Red ($r_t < 1$) indicates prompt-only has lower error. Two consistent patterns appear: on the left of the black line (prompt tokens), rows skew slightly red, reflecting that prompt-only better preserves prompt activations. On the right (decode tokens), the plot turns predominantly blue, showing that RAC substantially reduces reconstruction error exactly where long reasoning chains occur during autoregressive decoding. This demonstrates that by reconstructing CoT activations during calibration, RAC simulates the inference-time distribution and lowers decoding error on held-out test problems.

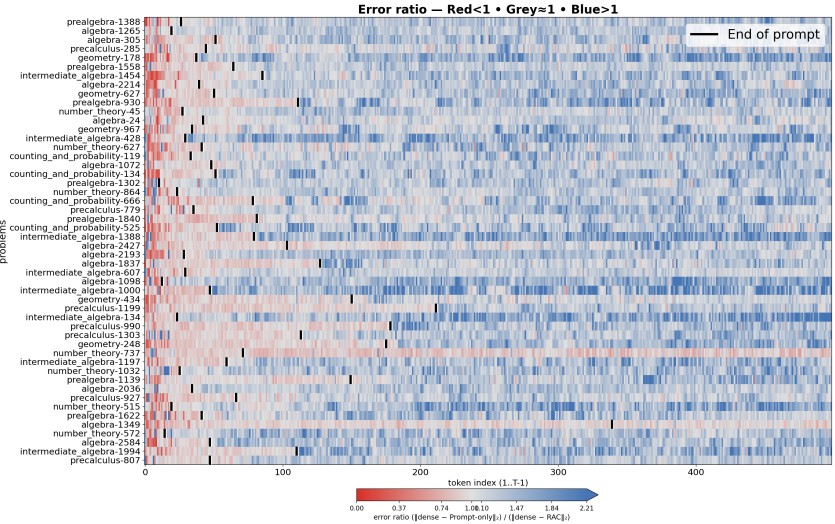

Figure 2: Tokenwise reconstruction error ratio on MATH500 test problems for pruned DeepSeek-R1-Distill-7B. Each row is a held-out problem, columns are token indices, and the black vertical line marks the prompt/decoding boundary. Colors show the ratio $r_t = e_t^{(\text{Prompt})}/e_t^{(\text{RAC})}$: grey $\approx 1$ indicates equal error, red $< 1$ indicates lower error from prompt-only calibration, and blue $> 1$ indicates lower error from RAC. Prompt-only slightly outperforms on the input tokens, but RAC has smaller error throughout the much longer decode phase.

## 5.2 ADDITIONAL EXPERIMENTS

**Throughput Analysis with Structured Pruning and Quantization (Table 6).** Moving beyond unstructured pruning, we evaluate semi-structured 2:4 sparsity patterns that remove exactly 2 out of every 4 contiguous weights, which can achieve actual speedups on modern hardware platforms such as NVIDIA Ampere (Mishra et al., 2021). The results demonstrate that RAC maintains its effectiveness even with structured pruning constraints. Notably, when applied to different portions of the model (first third, first and last third, or entire model), as is done in SparseGPT (Frantar & Alistarh, 2023), RAC consistently preserves accuracy while achieving meaningful throughput improvements. The combination of RAC with FP8 quantization (RAC+FP8) shows particularly promising results, achieving both high accuracy (0.940 for first and last third pruning) and substantial throughput gains (1675 tok/s vs 1426 tok/s baseline).

**On-policy vs. Off-policy Calibration (Table 7).** We investigate whether the CoT traces used during calibration must come from the same model being compressed (on-policy) or can be generated by a different model (off-policy). Specifically, we compare using traces from the 7B model itself versus using traces generated by the larger 14B model when compressing the 7B model. The results show that on-policy calibration generally outperforms off-policy calibration, particularly at higher sparsity levels (50% sparsity: 0.900 vs 0.876 accuracy). This suggests that the activation patterns during CoT generation are model-specific, and using traces from a different model creates a distribution mismatch. However, the performance gap is relatively modest at lower sparsity levels, indicating some robustness to off-policy data when compression is less aggressive.

**Run-time at smaller maximum test-time decoding length (Table 8).** To investigate whether the runtime penalties observed with pruned reasoning models are inherent to compression or primarily driven by excessive token generation, we evaluated both dense and RAC-pruned 7B models on MATH-500 under varying maximum decoding constraints of 32,768, 8,192, and 4,096 tokens. The runtime analysis reveals that the problematic behavior of pruned reasoning models is primarily confined to scenarios with excessive token budgets. At 32,768 tokens, the RAC-pruned model (40% sparsity) exhibits the pattern documented throughout this work, maintaining reasonable accuracy (91.2% versus 93.6% for the dense model) but requiring longer inference time (29.1 versus 23.3 minutes) due to longer, more rambling CoT traces. However, this runtime penalty largely disappears

at more practical token limits. At 8,192 tokens, the RAC-pruned model achieves nearly identical performance to the dense baseline on both accuracy (89.8% versus 90.4%) and runtime (10.4 versus 10.5 minutes). At 4,096 tokens, the pruned model actually outperforms the dense model on accuracy (83.6% versus 82.4%) while maintaining comparable runtime (8.6 versus 8.4 minutes). These results suggest that the runtime overhead from pruning manifests primarily when models are permitted to generate excessively long outputs, and that RAC-compressed models behave much more similarly to their dense counterparts when constrained to reasonable decoding lengths.

**Varying the pruning algorithm (Table 9).** In preceding sections we use SparseGPT as the default pruning baseline when contrasting prompt-only versus RAC calibration. As shown in Table 9, RAC can also be used in conjunction with other LLM pruning algorithms, such as ALPS and WANDA, and provides gains that are agnostic to the pruning algorithm. Comparing against the SparseGPT results in Table 2, we also find that ALPS is a competitive (and sometimes stronger) pruning baseline in its own right: for example, on Qwen3-1.7B at 50% sparsity, ALPS+RAC attains $0.788$ MATH500 accuracy versus $0.648$ for SparseGPT+RAC, and on Qwen3-8B at 50% sparsity ALPS+RAC reaches $0.940$ compared to $0.862$ for SparseGPT+RAC. For completeness, we report AIME-25 benchmark results for ALPS in Table 10. We observe similar trends as discussed above.

## 6 CONCLUSION AND LIMITATIONS

**Conclusion.** We studied the challenge of compressing reasoning language models, where inference is dominated by long chains of self-generated tokens. Standard pruning and quantization pipelines, which calibrate only on prompt activations, strongly degrade accuracy and runtime in this setting, often producing longer and less reliable reasoning traces. We proposed *Reasoning-Aware Compression* (RAC), a simple modification that augments calibration with on-policy CoT activations. RAC integrates seamlessly with existing pruning algorithms such as SparseGPT, requiring no retraining or distillation. Our experiments on mathematics and coding benchmarks demonstrate that RAC substantially mitigates the accuracy loss of pruning and stabilizes CoT generation.

**Limitations.** While RAC provides a simple and effective drop-in improvement, several limitations remain. First, this work does not directly address inference runtime increases. RAC improves pruning quality, which indirectly reduces unnecessary decoding steps, but designing compression methods that explicitly optimize to avoid sequence length increases remains an open direction. We show that this can be mitigated by simply reducing the maximum decoding budget, which is typically excessively high. Second, collecting on-policy CoT activations increases calibration cost relative to prompt-only pipelines, especially for large decode budgets, though this overhead is very modest compared to training or distillation.

## 7 ACKNOWLEDGEMENTS

Ryan Lucas contributed to this work while he was an intern at LinkedIn during summer 2025. This work is not a part of his MIT research. Rahul Mazumder contributed to this work while he was a consultant for LinkedIn (in compliance with MIT's outside professional activities policies). This work is not a part of his MIT research.

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

# 8 APPENDIX

Table 6: Actual throughput gains and accuracy with semi-structured 2:4 sparsity of MLP layers. Pruned model is Deepseek-R1-Distill-14B on MATH-500. Each pruning scope has two columns: **Accuracy (acc@1:1 (SE))** and **throughput (tok/s)**. Dense baseline throughput = 1426 tok/s.

| | First third | | First & last third | | Entire Model | |
|---|---|---|---|---|---|---|
| | Acc | Thr | Acc | Thr | Acc | Thr |
| RAC | 0.950 (0.001) | **1532** | **0.940** (0.012) | 1561 | **0.896** (0.014) | 1740 |
| Prompt+FP8 | **0.952** (0.010) | **1532** | 0.904 (0.0132) | **1675** | — | — |
| RAC+FP8 | 0.950 (0.010) | **1532** | 0.920 (0.0106) | **1675** | 0.8920 (0.014) | **1823** |
| Dense baseline | | | Throughput = 1426 tok/s | | | |

Table 7: On-policy (RAC) vs. off-policy calibration when pruning DeepSeek-R1-Distill-**7B** using DeepSeek-R1-Distill-**14B** traces on MATH500. We report **acc@1:1** with standard error.

| Sparsity | On-policy | Off-policy |
|---|---|---|
| 20% | **0.934** (0.011) | 0.930 (0.0114) |
| 30% | **0.934** (0.011) | 0.930 (0.0114 |
| 40% | 0.912 (0.013) | **0.916** (0.0124) |
| 50% | **0.900** (0.013) | 0.876 (0.0148) |

Table 8: MATH500 accuracy and runtime (minutes) across different **test-time maximum decode budgets** (number of tokens) for 7B models.

| Model | Max tokens = 32,768 | | Max tokens = 8,192 | | Max tokens = 4,096 | |
|---|---|---|---|---|---|---|
| | Acc. | Time | Acc. | Time | Acc. | Time |
| Dense 7B | **0.936** | **23.3** | **0.904** | 10.5 | 0.824 | **8.4** |
| Pruned 7B (RAC, 40%) | 0.912 | 29.1 | 0.898 | **10.4** | **0.836** | 8.6 |

Table 9: MATH500 accuracy $\pm$ SE under one-shot pruning for ALPS vs. WANDA for the Qwen-3 model family. Best accuracy in each row in **green**.

| Model | Sparsity | ALPS | | | WANDA | | |
|---|---|---|---|---|---|---|---|
| | | C4 | Prompt-Only | RAC | C4 | Prompt-Only | RAC |
| Qwen3-1.7B | *Dense* | **0.906 (0.013)** | **0.906 (0.013)** | **0.906 (0.013)** | **0.906 (0.013)** | **0.906 (0.013)** | **0.906 (0.013)** |
| | 30% | 0.796 (0.018) | **0.866 (0.015)** | 0.860 (0.016) | 0.774 (0.019) | 0.820 (0.017) | 0.828 (0.017) |
| | 40% | – | 0.812 (0.018) | **0.854 (0.016)** | 0.350 (0.021) | 0.706 (0.020) | 0.746 (0.020) |
| | 50% | – | 0.566 (0.022) | **0.788 (0.018)** | – | – | 0.436 (0.022) |
| Qwen3-8B | *Dense* | **0.962 (0.009)** | **0.962 (0.009)** | **0.962 (0.009)** | **0.962 (0.009)** | **0.962 (0.009)** | **0.962 (0.009)** |
| | 30% | 0.956 (0.009) | 0.956 (0.009) | **0.966 (0.008)** | 0.958 (0.009) | **0.966 (0.008)** | 0.960 (0.009) |
| | 40% | 0.902 (0.013) | 0.954 (0.009) | **0.962 (0.009)** | 0.936 (0.011) | 0.928 (0.012) | 0.950 (0.010) |
| | 50% | – | 0.890 (0.014) | **0.940 (0.011)** | – | 0.814 (0.017) | 0.878 (0.015) |
| Qwen3-14B | *Dense* | **0.972 (0.007)** | **0.972 (0.007)** | **0.972 (0.007)** | **0.972 (0.007)** | **0.972 (0.007)** | **0.972 (0.007)** |
| | 30% | 0.952 (0.010) | 0.962 (0.009) | 0.960 (0.009) | 0.966 (0.008) | 0.962 (0.009) | **0.968 (0.008)** |
| | 40% | 0.950 (0.010) | 0.958 (0.007) | **0.962 (0.009)** | 0.948 (0.010) | 0.958 (0.009) | **0.962 (0.009)** |
| | 50% | 0.822 (0.017) | 0.934 (0.011) | **0.960 (0.009)** | 0.870 (0.015) | 0.926 (0.012) | 0.930 (0.011) |

Table 10: Qwen3 AIME-25 accuracy under one-shot pruning with ALPS. Accuracy with standard error (SE) in parentheses. Best accuracy in each row in **green**.

| Model | Sparsity | C4 | Prompt-Only | RAC |
|---|---|---|---|---|
| Qwen3-1.7B | *Dense* | **0.333 (0.088)** | **0.333 (0.088)** | **0.333 (0.088)** |
| | 30% | 0.233 (0.079) | 0.267 (0.082) | **0.400 (0.091)** |
| | 40% | 0.033 (0.033) | 0.267 (0.082) | **0.433 (0.092)** |
| | 50% | 0.000 (0.000) | 0.033 (0.033) | **0.267 (0.082)** |
| Qwen3-8B | *Dense* | **0.633 (0.090)** | **0.633 (0.090)** | **0.633 (0.090)** |
| | 30% | 0.567 (0.092) | 0.567 (0.092) | **0.700 (0.085)** |
| | 40% | 0.433 (0.092) | 0.500 (0.093) | **0.667 (0.087)** |
| | 50% | 0.033 (0.033) | 0.367 (0.089) | **0.467 (0.093)** |
| Qwen3-14B | *Dense* | **0.667 (0.088)** | **0.667 (0.088)** | **0.667 (0.088)** |
| | 30% | **0.733 (0.082)** | 0.600 (0.091) | 0.633 (0.089) |
| | 40% | 0.533 (0.093) | 0.667 (0.087) | **0.700 (0.085)** |
| | 50% | 0.267 (0.082) | 0.533 (0.093) | **0.667 (0.087)** |

