# OpenReview forum: "Reasoning Models Can be Accurately Pruned Via Chain-of-Thought Reconstruction"
_ICLR.cc/2026/Conference — ICLR 2026 Poster_

### Official Review · Reviewer_1Hgg · 2025-10-27

**Soundness:** 3
**Presentation:** 3
**Contribution:** 2
**Rating:** 6
**Confidence:** 2

**Summary:**

This paper proposes a pruning strategy for reasoning models. Specifically, it uses the fact that there are more decoding tokens than input tokens for reasoning models, so it collects activations for those and applies existing pruning methods.

**Strengths:**

- The paper is easy to read and follow.
- The solution is simple and straightforward.
- The results show that it preserves accuracy and improves efficiency.

**Weaknesses:**

- Isn’t the self-calibration a somewhat similar mechanism [1]? Please correct me if I’m wrong.
- Missing comparison with efficient reasoning on the token side.
    - What is the pruning cost compared to the fine-tuning cost of training the model to perform concise reasoning?
    - How is model pruning more effective compared to token reduction? Reducing decoding tokens might reduce the cost more. Also, since both can be applied, I wonder what would happen if we applied both.
- Runtime minutes appear reversed. It seems like the dense model is the most efficient, and pruning makes inference slower according to the tables.
- Evaluation sets for math are limited to MATH500, and the models already perform very well on it (90%+ for 7B+ models). It is not surprising that pruning preserves high accuracy. I would like to see results on AIME as well.

[1] Williams et al. “Self-calibration for Language Model Quantization and Pruning”, NACCL 2025

**Questions:**

- Are the runtime minutes reversed, or am I misunderstanding something?
- As in the weaknesses, comparing with efficient fine-tuning for short CoT would make the paper more valuable.
- Why do you think naive pruning leads to more tokens in the output?

---

> ### Author Response · Authors · 2025-11-20
>
> We thank the reviewer for their thoughtful comments and suggestions. Below, we present additional results and clarifications requested by the reviewer. We hope our responses address their concerns.
>
> > **Isn’t the self-calibration a somewhat similar mechanism [1]? Please correct me if I’m wrong.**
>
> Thank you for bringing this paper to our attention. We have included this reference in Section 3, “Calibration data for post-training pruning” of the revised manuscript. While the mechanism is superficially similar (both methods use model-generated text as calibration data), the implementation, motivation and use cases are different in our work. In [1], self-calibration is explicitly based on free-form text generation: they start from a bare Beginning-of-Sequence (BOS) token and let the base LM generate until EOS or a maximum length is reached and never condition on task prompts, i.e.:
>
> $z_{0:T} \sim p_{\theta} (z),  z_0 = \text{BOS}$
>
> and they solve the standard layerwise objective
>
> $$ \min_{\widehat{\mathbf W}}
> 𝔼 [|| (\mathbf W - \widehat{\mathbf W})\mathbf{X} (z) ]
> $$
>
>
> Where $\mathbf{X} (z)$ are the activations corresponding to $z$. We remark that, under this scheme, $p_{\theta}$ will in general approximate the (pre-)training text distribution of the base model. This matches the stated goal of [1] to generate a calibration set that resembles the model’s training dataset, which might not be publicly available. By contrast, RAC does not aim to recover the training distribution: we condition on task prompts and use the model’s own long chain-of-thought rollouts, so our calibration activations are matched to the model’s inference-time behaviour on reasoning tasks rather than to generic free-form text. Finally, we note that [1] does not consider difficult math and coding reasoning benchmarks as we do.
>
> [1] Self-calibration for Language Model Quantization and Pruning, Miles Williams, George Chrysostomou, Nikolaos Aletras

---

> ### Author Response · Authors · 2025-11-20
>
> > **Missing comparison with efficient reasoning on the token side. What is the pruning cost compared to the fine-tuning cost of training the model to perform concise reasoning?**
>
> Thank you for your question. Pruning is in general much less expensive than fine-tuning a model to do concise reasoning. For example, we can perform one-shot pruning on a 32B parameter reasoning model end-to-end on a single GPU. In contrast, a popular method L1 [2] for fine-tuning reasoning models to have more concise CoT can require a cluster of 8 GPUs e.g., see the default implementation of L1 from their official codebase:
>
> https://github.com/cmu-l3/l1/blob/main/scripts/train/run_l1_exact.sh
>
> Moreover, you may notice that the main experiments in the L1 paper are performed on DeepSeek's 1.5B parameter model, while they have some additional experiments called "Scaling to Larger Models" where they do limited experiments on a 7B parameter model, but this presumably requires even more resources. In contrast, as we show in Table 6 of the paper, we are able to achieve meaningful throughput lift via cheap one-shot pruning and little loss of accuracy.
>
> In general, as you allude to, we believe that these methods can be used in conjunction with pruning to lead to models that (1) are compressed on the weights side and (2) emit only as many tokens as necessary. This can lead to extra acceleration since it would provide both faster forward passes and fewer forward passes owing to (1) and (2) respectively.
>
> > **Runtime minutes appear reversed. It seems like the dense model is the most efficient, and pruning makes inference slower according to the tables.**
>
> The runtime minutes are not reversed. In general, we find that especially with naive pruning (e.g. on C4 calibration set) that pruned models exhibit counterintuitive run-time increases, which is attributable to more generated tokens. For example, when pruning the DeepSeek-R1-Distill-7B model with a C4 calibration set, we observe that the model can senselessly start repeating itself. We show an example of a simple question and output below:
>
> ```
> Prompt: If a bookstore sells 200 books in one day, which is 25% more than the previous day, how many books did it sell the previous day?
>
> Alright, let me think step by step...
>
> First, I need to figure out how much more the current day is compared to the previous day. So, 25% more means that the current day is 25% higher. So, the previous day is 200, and the current day is 25% higher. So, I need to find out the previous day's sale.
>
> So, the current day is 200, which is 25% more than the previous day. So, the previous day is 200, and the current day is 25% more. So, I need to find out the previous day's sale.
>
> So, the current day is 200, which is 25% more than the previous day. So, the previous day is 200, and the current day is 25% more. So, I need to find out the previous day's sale.
>
> So, the current day is 200, which is 25% more than the previous day. So, the previous day is 200, and the current day is 25% more. So, I need to find out the previous day's sale.
> ```
>
> As to why this occurs, it is not entirely clear to us. One hypothesis is that since the dense reasoning model has been fine-tuned on sequences that generally include the end-of-sequence and answer tokens, when the model is pruned on calibration data that never includes these tokens, it can proportionally downweight them for sampling. Similarly, since the data also contains no examples of reasoning, the model may also forget how to backtrack effectively or reason entirely, as is the case in the above example. The result is more generated tokens and therefore longer run-times, even though the underlying network is more sparse. This might explain why we do not see such behavior with our RAC framework.
>
> As we noted in the limitations and as you point out, there is motivation to implement efficient reasoning on the token side with efficient reasoning on the model side (i.e., pruning) to correct this issue. However, we show in our additional experiments (e.g., Table 8 of the revised manuscript) that this problem can be alleviated to a large extent by just reducing the test-time decode budget. This forces the model to provide an answer early, and once this is implemented the pruned model typically produces a similar number of tokens to the dense model while also being capable of being accelerated via sparsity
>
>
> [2] L1: Controlling How Long A Reasoning Model Thinks With Reinforcement Learning, Pranjal Aggarwal, Sean Welleck

---

> > ### Author Response · Authors · 2025-11-20
> >
> > > **Evaluation sets for math are limited to MATH500, and the models already perform very well on it (90%+ for 7B+ models). It is not surprising that pruning preserves high accuracy. I would like to see results on AIME as well.**
> >
> > Thank you for your comment. Under the same experimental setup as with MATH500, we have performed additional experiments on the Qwen3 family. We report the benchmark results on the AIME25 dataset below, when pruning is done via either SparseGPT or Wanda. The results have also been added to the paper (Tables 4 and 10).
> >
> > ### Qwen3 AIME-25 pass@1 accuracy ($\pm$ SE) under one-shot pruning with SparseGPT
> >
> >
> >
> >
> > | Model | Sparsity | C4 | Prompt-Only | RAC |
> > |-------|----------|-----------|--------------|------------|
> > | **Qwen3-1.7B** | *Dense* | **0.333 (0.088)** | **0.333 (0.088)** | **0.333 (0.088)** |
> > |  | 30% | 0.267 (0.082) | **0.300 (0.085)** | 0.267 (0.082) |
> > |  | 40% | 0.000 (0.000) | 0.133 (0.063) | **0.267 (0.082)** |
> > |  | 50% | 0.000 (0.000) | 0.000 (0.000) | **0.200 (0.074)** |
> > | **Qwen3-8B** | *Dense* | **0.633 (0.090)** | **0.633 (0.090)** | **0.633 (0.090)** |
> > |  | 30% | 0.600 (0.091) | 0.633 (0.090) | **0.667 (0.088)** |
> > |  | 40% | 0.533 (0.093) | **0.633 (0.090)** | **0.633 (0.090)** |
> > |  | 50% | 0.033 (0.033) | 0.300 (0.085) | **0.500 (0.093)** |
> > | **Qwen3-14B** | *Dense* | **0.667 (0.088)** | **0.667 (0.088)** | **0.667 (0.088)** |
> > |  | 30% | **0.700 (0.085)** | 0.667 (0.088) | 0.667 (0.088) |
> > |  | 40% | 0.500 (0.093) | 0.633 (0.090) | **0.667 (0.088)** |
> > |  | 50% | 0.300 (0.085) | 0.433 (0.092) | **0.600 (0.091)** |
> >
> >
> >
> >
> > ### Qwen3 AIME-25 pass@1 accuracy ($\pm$ SE) under one-shot pruning with ALPS
> >
> >
> > | Model | Sparsity | C4 | Prompt-Only | RAC |
> > |-------|----------|-------------|----------------|--------------|
> > | **Qwen3-1.7B** | *Dense* | **0.333 (0.088)** | **0.333 (0.088)** | **0.333 (0.088)** |
> > |  | 30% | 0.233 (0.079) | 0.267 (0.082) | **0.400 (0.091)** |
> > |  | 40% | 0.033 (0.033) | 0.267 (0.082) | **0.433 (0.092)** |
> > |  | 50% | 0.000 (0.000) | 0.033 (0.033) | **0.267 (0.082)** |
> > | **Qwen3-8B** | *Dense* | **0.633 (0.090)** | **0.633 (0.090)** | **0.633 (0.090)** |
> > |  | 30% | 0.567 (0.092) | 0.567 (0.092) | **0.700 (0.085)** |
> > |  | 40% | 0.433 (0.092) | 0.500 (0.093) | **0.667 (0.087)** |
> > |  | 50% | 0.033 (0.033) | 0.367 (0.089) | **0.467 (0.093)** |
> > | **Qwen3-14B** | *Dense* | **0.667 (0.088)** | **0.667 (0.088)** | **0.667 (0.088)** |
> > |  | 30% | **0.733 (0.082)** | 0.600 (0.091) | 0.633 (0.089) |
> > |  | 40% | 0.533 (0.093) | 0.667 (0.087) | **0.700 (0.085)** |
> > |  | 50% | 0.267 (0.082) | 0.533 (0.093) | **0.667 (0.087)** |
> >
> >
> > We see that RAC preserves the quality better than other baselines, similar to the MATH500 case.
> >
> > We thank the reviewer again for their helpful comments and hope our response and new results have addressed the comment you raised.

---

### Official Review · Reviewer_LocM · 2025-10-27

**Soundness:** 3
**Presentation:** 2
**Contribution:** 3
**Rating:** 4
**Confidence:** 3

**Summary:**

This paper identifies that pruning reasoning models (e.g., DeepSeek-R1) with standard methods fails, causing accuracy drops and slower inference. The authors blame a calibration mismatch: methods use prompt activations, but reasoning is decode-dominated. The fix, Reasoning-Aware Compression (RAC), adds on-policy Chain-of-Thought (CoT) activations to the calibration set. Experiments show RAC significantly boosts accuracy (e.g., 90.0% vs. 74.4% at 50% sparsity for a 7B model) and reduces the "rambling" CoTs that slow down inference.

**Strengths:**

1. Clear Problem & Diagnosis: Clearly identifies and diagnoses a non-obvious problem: pruning reasoning models can slow them down due to a prompt/decode activation mismatch (Fig. 1).

2. Simple & Practical Solution: The proposed solution (RAC) is simple and practical: a "drop-in" fix for the calibration data, not a complex new algorithm.

3. Strong Validation (on DeepSeek): Provides strong empirical validation on the DeepSeek-R1 family, showing consistent gains on math/code and a clear reason for success (lower decode error in Fig. 2).

**Weaknesses:**

1. Generalizability Concerns: I think this is the primary weakness. The claims are broad, but results are confined only to the DeepSeek-R1 (GRPO-trained) family. There is no evidence this works on standard models (Llama, Mistral) using CoT prompting or other reasoning models like Qwen3, QwQ. This severely limits the contribution's impact.

2. Incomplete Method Comparison: The comparison of pruning methods is incomplete (Table 6). To validate RAC as algorithm-agnostic, the baseline (C4) results for ALPS and Wanda are needed to show relative lift, not just absolute performance.

3. Unquantified Overhead: The paper notes RAC adds overhead but never quantifies the wall-clock time for calibration versus the baseline, making its practicality hard to assess.

**Questions:**

1. Will RAC generalize to standard models (Llama, Mistral) using CoT prompting or other reasoning models like Qwen3, QwQ, or is the benefit specific to GRPO-trained models like DeepSeek-R1? And I think I may improve my score if some of the experiments can be done.

2. Please quantify the calibration overhead (wall-clock time) of RAC versus the 1M C4 baseline.

3. Table 4 shows on-policy traces are best. Is matching a model's idiosyncratic activation patterns more important than using 'cleaner' traces from a larger model?

---

> ### Author Response · Authors · 2025-11-20
>
> We thank the reviewer for providing us with helpful feedback and suggestions. Below, we present additional results requested by the reviewer. We hope our responses address their concerns.
>
> > **Generalizability Concerns: I think this is the primary weakness. The claims are broad, but results are confined only to the DeepSeek-R1 (GRPO-trained) family. There is no evidence this works on standard models (Llama, Mistral) using CoT prompting or other reasoning models like Qwen3, QwQ. This severely limits the contribution's impact.**
>
> We thank the reviewer for this comment. We have run extra experiments on the Qwen3 reasoning model family and LLama reasoning model family. We have shown results for pruning Qwen3 (1.7B, 8B, 14B) and LLama (8B, 70B) to sparsity of 30%, 40%, and 50%. We present evaluation results on MATH-500 and on AIME (at the request of reviewer 1Hgg). The results show very similar findings to the DeepSeek-R1 variants. RAC preserves reasoning performance on mathematical reasoning much better than C4 and prompt-only datasets, while reducing the tendency of the model to degenerate and produce long sequences. We have incorporated these changes in the paper and present the summary of our results here for completeness.  We hope this can strengthen the claims of generalizability of our method. We included versions of these tables with evaluation runtimes in the revised paper (we refer to Table 2 for Qwen 3 and MATH500, Table 3 for Llama and MATH500, and Table 4 for Qwen 3 and AIME25).
>
> ### Qwen3 MATH500 accuracy ($\pm$ SE) under one-shot pruning with SparseGPT
> | Model | Sparsity | C4 | Prompt-Only | RAC |
> |-------|----------|-----------|--------------|------------|
> | **Qwen3-1.7B** | *Dense* | **0.906 (0.013)** | **0.906 (0.013)** | **0.906 (0.013)** |
> |  | 30% | 0.822 (0.017) | 0.874 (0.015) | **0.880 (0.015)** |
> |  | 40% | 0.346 (0.021) | 0.764 (0.019) | **0.858 (0.016)** |
> |  | 50% | — | 0.470 (0.022) | **0.648 (0.021)** |
> | **Qwen3-8B** | *Dense* | **0.962 (0.009)** | **0.962 (0.009)** | **0.962 (0.009)** |
> |  | 30% | 0.948 (0.010) | 0.958 (0.009) | **0.972 (0.007)** |
> |  | 40% | 0.906 (0.013) | 0.944 (0.010) | **0.968 (0.008)** |
> |  | 50% | 0.564 (0.022) | 0.470 (0.022) | **0.862 (0.015)** |
> | **Qwen3-14B** | *Dense* | **0.972 (0.007)** | **0.972 (0.007)** | **0.972 (0.007)** |
> |  | 30% | 0.950 (0.010) | 0.964 (0.008) | **0.970 (0.008)** |
> |  | 40% | 0.958 (0.009) | 0.962 (0.009) | **0.970 (0.008)** |
> |  | 50% | 0.830 (0.017) | 0.932 (0.011) | **0.962 (0.009)** |
>
> ### Qwen3 AIME-25 pass@1 accuracy ($\pm$ SE) under one-shot pruning with SparseGPT
>
>
>
>
> | Model | Sparsity | C4 | Prompt-Only | RAC |
> |-------|----------|-----------|--------------|------------|
> | **Qwen3-1.7B** | *Dense* | **0.333 (0.088)** | **0.333 (0.088)** | **0.333 (0.088)** |
> |  | 30% | 0.267 (0.082) | **0.300 (0.085)** | 0.267 (0.082) |
> |  | 40% | 0.000 (0.000) | 0.133 (0.063) | **0.267 (0.082)** |
> |  | 50% | 0.000 (0.000) | 0.000 (0.000) | **0.200 (0.074)** |
> | **Qwen3-8B** | *Dense* | **0.633 (0.090)** | **0.633 (0.090)** | **0.633 (0.090)** |
> |  | 30% | 0.600 (0.091) | 0.633 (0.090) | **0.667 (0.088)** |
> |  | 40% | 0.533 (0.093) | **0.633 (0.090)** | **0.633 (0.090)** |
> |  | 50% | 0.033 (0.033) | 0.300 (0.085) | **0.500 (0.093)** |
> | **Qwen3-14B** | *Dense* | **0.667 (0.088)** | **0.667 (0.088)** | **0.667 (0.088)** |
> |  | 30% | **0.700 (0.085)** | 0.667 (0.088) | 0.667 (0.088) |
> |  | 40% | 0.500 (0.093) | 0.633 (0.090) | **0.667 (0.088)** |
> |  | 50% | 0.300 (0.085) | 0.433 (0.092) | **0.600 (0.091)** |
>
>
> ### Distill-R1 Llama MATH500 accuracy ($\pm$ SE) under one-shot pruning with SparseGPT
>
>
> | Model | Sparsity | C4 | Prompt-Only | RAC |
> |-------|----------|-----------|--------------|------------|
> | **LLaMA-8B** | *Dense* | **0.866 (0.015)** | **0.866 (0.015)** | **0.866 (0.015)** |
> |  | 30% | 0.860 (0.016) | 0.850 (0.016) | **0.884 (0.014)** |
> |  | 40% | 0.794 (0.018) | 0.804 (0.018) | **0.834 (0.017)** |
> |  | 50% | 0.508 (0.022) | 0.654 (0.021) | **0.714 (0.020)** |
> | **LLaMA-70B** | *Dense* | **0.954 (0.009)** | **0.954 (0.009)** | **0.954 (0.009)** |
> |  | 30% | 0.936 (0.011) | **0.946 (0.010)** | **0.946 (0.010)** |
> |  | 40% | 0.910 (0.013) | **0.936 (0.011)** | 0.932 (0.011) |
> |  | 50% | 0.818 (0.017) | 0.892 (0.014) | **0.904 (0.013)** |

---

> > ### Author Response · Authors · 2025-11-20
> >
> > > **Incomplete Method Comparison: The comparison of pruning methods is incomplete (Table 6). To validate RAC as algorithm-agnostic, the baseline (C4) results for ALPS and Wanda are needed to show relative lift, not just absolute performance.**
> >
> > To address your concerns, we have performed additional relative comparisons with two other algorithms, ALPS and WANDA, on the same Qwen3 model architectures. We find that RAC provides a performance gain to both ALPS and WANDA at various sparsity levels. The results are summarized below.
> >
> >
> >
> > ### MATH500 accuracy $\pm$ SE under one-shot pruning for ALPS for the Qwen-3 model family.
> > | Model | Sparsity | C4 | Prompt-Only | RAC |
> > |-------|----------|-----------|---------------------|-------------|
> > | **1.7B** | *Dense* | **0.906 (0.013)** | **0.906 (0.013)** | **0.906 (0.013)** |
> > |  | 30% | 0.796 (0.018) | **0.866 (0.015)** | 0.860 (0.016) |
> > |  | 40% | — | 0.812 (0.018) | **0.854 (0.016)** |
> > |  | 50% | — | 0.566 (0.022) | **0.788 (0.018)** |
> > | **8B** | *Dense* | **0.962 (0.009)** | **0.962 (0.009)** | **0.962 (0.009)** |
> > |  | 30% | 0.956 (0.009) | 0.956 (0.009) | **0.966 (0.008)** |
> > |  | 40% | 0.902 (0.013) | 0.954 (0.009) | **0.962 (0.009)** |
> > |  | 50% | — | 0.890 (0.014) | **0.940 (0.011)** |
> > | **14B** | *Dense* | **0.972 (0.007)** | **0.972 (0.007)** | **0.972 (0.007)** |
> > |  | 30% | 0.952 (0.010) | 0.962 (0.009) | 0.960 (0.009) |
> > |  | 40% | 0.950 (0.010) | 0.958 (0.007) | **0.962 (0.009)** |
> > |  | 50% | 0.822 (0.017) | 0.934 (0.011) | **0.960 (0.009)** |
> >
> >
> > The following table summarizes the results for the Wanda pruning algorithm.
> >
> >
> > ### MATH500 accuracy $\pm$ SE under one-shot pruning for WANDA for the Qwen-3 model family.
> > | Model | Sparsity  | C4 | Prompt-Only | RAC |
> > |-------|-------------|--------------|-----------------------|---------------|
> > | **1.7B** | *Dense* | **0.906 (0.013)** | **0.906 (0.013)** | **0.906 (0.013)** |
> > |  | 30% |  0.774 (0.019) | 0.820 (0.017) | **0.828 (0.017)** |
> > |  | 40% | — |  0.706 (0.020) | **0.746 (0.020)** |
> > |  | 50%  | — | — | **0.436 (0.022)** |
> > | **8B** | *Dense* | **0.962 (0.009)** | **0.962 (0.009)** | **0.962 (0.009)** |
> > |  | 30% | 0.958 (0.009) | **0.966 (0.008)** | 0.960 (0.009) |
> > |  | 40% |  0.936 (0.011) | 0.928 (0.012) | **0.950 (0.010)** |
> > |  | 50% | — | 0.814 (0.017) | **0.878 (0.015)** |
> > | **14B** | *Dense* | **0.972 (0.007)** | **0.972 (0.007)** | **0.972 (0.007)** |
> > |  | 30% | 0.966 (0.008) | 0.962 (0.009) | **0.968 (0.008)** |
> > |  | 40% | 0.948 (0.010) | 0.958 (0.009) | **0.962 (0.009)** |
> > |  | 50% | 0.870 (0.015) | 0.926 (0.012) | **0.930 (0.011)** |
> >
> > We have included these new experiments in the revised paper submission (Table 9).

---

> > > ### Author Response · Authors · 2025-11-20
> > >
> > > > **Unquantified Overhead: The paper notes RAC adds overhead but never quantifies the wall-clock time for calibration versus the baseline, making its practicality hard to assess.**
> > >
> > > We present below the run-times for collecting the on-policy RAC traces (1M calibration tokens) for Qwen3-8B and the time taken to prune the model with SparseGPT with a calibration dataset of size 1M (this is typically very similar no matter what data is in the calibration set). As for the overhead of our procedure, it is typically very modest compared to fine-tuning or distillation which typically require clusters of GPUs. In contrast we only require inference from the base model (first to sample the token, and then to compute activations), which can be done on a single GPU. For larger architectures, the overhead for RAC will scale with the inference speed of the base model, though we find we were able to run end-to-end pruning for all architectures we tested (even 32B, 70B) in under 24 hours on a single GPU. Additionally, we note that collecting the on-policy traces is a one-time cost, and the same calibration set can be reused to prune the model multiple times to varying sparsity levels.
> > >
> > > | Stage | Time (hours) |
> > > | ------------------------------- | ------------ |
> > > | Reasoning Activation Collection | 9.04 |
> > > | SparseGPT Pruning | 0.25 |
> > >
> > > > **Table 4 shows on-policy traces are best. Is matching a model's idiosyncratic activation patterns more important than using 'cleaner' traces from a larger model?**
> > >
> > > Thank you for your thought-provoking question! We think matching the model’s activation distribution is important and can be more beneficial compared to using the traces from a bigger model. However, that is assuming the model has an overall good performance on the task. Otherwise, the reasoning traces from the model will not be informative (e.g., the model might just repeat itself) and we do not expect using those traces for pruning will be helpful. That being said, for such a low-quality model, we do not expect pruning to work well given pruning tries to match the dense model’s output.
> > > One intuition for why this might be can be drawn from on-policy distillation in reinforcement learning. [1] finds that, when performing distillation, it is often better to train a student model on its own rollouts, with a teacher providing feedback, rather than to train purely to match the “cleaner” teacher trajectories, because otherwise the student faces a distribution shift between training and inference time. In our setting, pruning on the model’s own CoT traces might play the same role: to ensure that the compressed model is optimized to reconstruct the idiosyncratic activation patterns it will actually visit at inference time, rather than those of a larger teacher whose trajectories it may not be expressive enough to match.
> > >
> > > We thank you for your detailed comments, and we hope our response addresses your concerns. If so, we would like to ask you to consider increasing the review score.
> > >
> > > [1] On-Policy Distillation of Language Models: Learning from Self-Generated Mistakes, Rishabh Agarwal, Nino Vieillard, Yongchao Zhou, Piotr Stanczyk, Sabela Ramos, Matthieu Geist, Olivier Bachem

---

### Official Review · Reviewer_hbbN · 2025-10-30

**Soundness:** 4
**Presentation:** 2
**Contribution:** 3
**Rating:** 4
**Confidence:** 3

**Summary:**

This is a method to compress models for reasoning. The key idea is that reasoning involves long generated sequences relatively to the prompts, compared to standard tasks / benchmark where the generated part is short or nonexistent for multi-choice. Hence, using the prompts themselves as "calibrating data" to prune is misguided and performance can be improved by using generated sequences as calibrating data. Experiments show that it works very well.

**Strengths:**

The method is simple, straightforward and comparisons to other methods from that domain demonstrate superior performance.

**Weaknesses:**

The proposed method is too simple and the contribution too limited. It boils down to saying "use the data of your task to calibrate pruning for your task".

Now, model compression is not a domain I know well and it is easy to have an illusion of triviality post-hoc.

About the form of the paper, some sections are just loading pages with equations which are in my opinion needless and give an impression of gratuitous filling to look impressive.

**Questions:**

I do not have question, but I am curious to see if other reviewers who know the field better confirm that this method is not already well known and used in practice.

---

> ### Author Response · Authors · 2025-11-20
>
> We thank the reviewer for their comments and questions. Below, we present our responses to the comments from the review.
>
> > **The proposed method is too simple and the contribution too limited. It boils down to saying "use the data of your task to calibrate pruning for your task"**
>
> Thank you for your comment. First, we would like to emphasize that our proposal is not to use the data of the task for calibration. Our proposal is to use the Chain-of-Thought (CoT) reasoning traces from the model itself to create the on-policy calibration set. In fact, as our experiments show (the “prompt-only”' option) that using the task data for calibration often leads to lower accuracy compared to our proposal (“RAC”).
>
> Secondly, unlike previous works that study the pruning of reasoning models, we show that it is possible to successfully prune reasoning models with little to no loss of accuracy, using our proposed RAC framework. Our solution might appear simple in hindsight (which we do not believe is a weakness), but it unlocks a new avenue for compressing reasoning models. For example,  the paper When Reasoning Meets Compression (Zhang et al., 2025) performs an extremely thorough study of pruning DeepSeek-R1-style models. They use SparseGPT for pruning but with a standard C4-style calibration pipeline. They never use on-policy CoT activations for calibration. They conclude that pruning reasoning models is extremely hard and usually hurts both accuracy and latency. For example, a direct quote from the paper is “pruning only 50% of the weights causes significant degradation, rendering the pruned Large Reasoning Models (LRMs) unusable.” Our method directly contradicts their negative conclusion using the exact same models and benchmarks. In addition, another concurrent submission to ours at ICLR 2026 entitled Self-Healing: Recovering Pruned Large Reasoning Models via Reinforcement Learning directly states: “These results indicate that without recovery training, existing pruning techniques cannot serve as effective compression methods for LRMs.” In contrast, our work demonstrates that with proper calibration that aligns with the inference-time distribution, one-shot pruning can effectively compress reasoning models without requiring expensive recovery training or distillation procedures.
>
> > **About the form of the paper, some sections are just loading pages with equations which are in my opinion needless and give an impression of gratuitous filling to look impressive.**
>
> We would like to clarify that though the core idea is simple, it is much better to be explicit in diagnosing the problem and providing a solution. What we convey in Eqs. 2-4 is what exactly the autoregressive process looks like (at the level of activations) for a reasoning LLM. We then relate that to what standard pruning objectives optimize to reveal the gap and finally which activations we collect and how they enter our pruning objective (Eqs 5-7). If we were vague about these details it would make it harder to understand why the method works and harder to reproduce for others in the community.

---

> ### Author Response · Authors · 2025-11-20
>
> > **I do not have question, but I am curious to see if other reviewers who know the field better confirm that this method is not already well known and used in practice.**
>
> As we discussed above, our proposed method has not appeared before, and it is an important development towards efficient reasoning LLMs. Since you admit you are not familiar with the LLM compression literature, we can provide you a brief summary with a contrast to our work. One-shot pruning of LLMs is a relatively recent field. Compared to pruning on e.g. traditional computer vision models, LLMs pose a unique challenge due to their scale and the nature of the (typically very large) training datasets. SparseGPT [1] showed that, surprisingly, by using a relatively small amount of standard C4 text data as a calibration set, LLMs can be pruned without large blow-ups in perplexity on other datasets. Thus, pipelines of this flavor have become standard, with various follow-up works such as WANDA [2] and ALPS [3] using similar C4-style calibration pipelines with good results for standard language modelling benchmarks. This has likewise been true in quantization [4], where C4 is the most common choice. There are some works such as [5] using self-generated tokens for compression, however they do free-form text generation starting from the Beginning-of-Sequence (BOS) token, with the stated intention of matching the model’s (pre-)training distribution in an unbiased way, as opposed to calibrating on a curated dataset like C4 which may introduce overfitting. However, this is quite distinct from our work where we focus on reasoning models and condition on task-specific problems for generation. We encourage you to engage with this literature and with the comments of the other reviewers, who have not raised novelty as an issue.
>
> Finally, in light of our clarifications, and overall support of other reviewers towards our work, we would like to ask you to consider increasing your review score.
>
> [1] SparseGPT: Massive Language Models Can Be Accurately Pruned in One-Shot, Elias Frantar, Dan Alistarh
>
> [2] A Simple and Effective Pruning Approach for Large Language Models, Mingjie Sun, Zhuang Liu, Anna Bair, J.Zico Kolter
>
> [3] ALPS: Improved Optimization for Highly Sparse One-Shot Pruning for Large Language Models, Xiang Meng, Kayhan Behdin, Haoyue Wang, Rahul Mazumder
>
> [4] AWQ: Activation-aware Weight Quantization for LLM Compression and Acceleration, Ji Lin, Jiaming Tang, Haotian Tang, Shang Yang, Wei-Ming Chen, Wei-Chen Wang, Guangxuan Xiao, Xingyu Dang, Chuang Gan, Song Han
>
> [5] Self-calibration for Language Model Quantization and Pruning, Miles Williams, George Chrysostomou, Nikolaos Aletras

---

### Official Review · Reviewer_bqFF · 2025-10-31

**Soundness:** 4
**Presentation:** 3
**Contribution:** 3
**Rating:** 6
**Confidence:** 3

**Summary:**

This paper introduces Reasoning-Aware Compression, a pruning method for reasoning language models that maintains high accuracy while improving efficiency. Unlike standard pruning, which calibrates using only prompt signals, RAC aligns pruning with decoding behavior by incorporating internal signals from both the prompt and the model’s own chain of thought. This alignment ensures pruning reflects how the model actually reasons during generation. Experiments show that RAC-pruned models produce shorter, cleaner reasoning chains and retain accuracy close to the dense baseline- a 7B model at 50% sparsity maintains about 90% accuracy, compared to 74.4% with standard calibration.

**Strengths:**

1. Aligns pruning with chain-of-thought activations, effectively preserving reasoning quality. The input column to be considered for pruning considers not just the provided input prompts tokens but aslo the LLM policy's self generated reasoning CoT tokens, which aligns the model with what it would do during inference time.

2. Achieves up to 95% dense-model accuracy at 50% sparsity with no retraining.

3. Simple, plug-and-play method compatible with existing SparseGPT or WanDA approaches.

**Weaknesses:**

1. How do you pre-calculate $\mathcal{D}_M$ which tells howmany times you would self-generate tokens from the model?
2. While the paper has good rigor the presentation and notations can be well-defined, especially the core part in page 4 and 5. Eg. what is $N_P$?

**Questions:**

1. Is there any study on how the pruning varies with the CoT quality. With long reasoning traces, many times we observe the model has a long approach then backtracks and then corrects itself etc to get the output. Did you observe any specific behaviors there?
2. How would this method generalize to other tasks?

---

> ### Author Response · Authors · 2025-11-20
>
> We would like to thank you for your positive feedback towards our manuscript. Below, we respond to your comments and questions.
>
> > **How do you pre-calculate $\mathcal{D}_M$ which tells how many times you would self-generate tokens from the model?**
>
> Thank you for your question and attention to detail. In practice, we do not fix $|D_M|$  directly. Instead, similar to other one-shot pruning papers, we fix a global calibration token budget $T$ and a per-sequence cap $T_{\max}$ and then sample until the budget is exhausted. Concretely, starting from an empty calibration set, we repeat the following procedure:
> 1. Draw a prompt $P_m$.
> 2. Run the dense model autoregressively to generate a decode sequence $D_m$ with length at most $T_{\max} = 8192$
> 3. Append the combined tokens $P_m \cup D_m$ to the
> calibration dataset
> 4. If the cumulative number of tokens has reached or exceeded the global
> budget $T = 1{,}000{,}000$, we stop. Otherwise, we return to step 1.
>
> Finally, if the last sequence overshoots the budget, we truncate the dataset to have a total length exactly $T$ tokens. Under this sampling scheme, the total number of decode tokens is then approximately
>
> $ |D_M| \approx T - \sum_{m=1}^M |P_m|, $
>
> up to the (typically small) truncation of the final sequence due to enforcing the global token budget $T$ and the per-sequence cap $T_{\max}$.
>
> > **While the paper has good rigor the presentation and notations can be well-defined, especially the core part in page 4 and 5. E.g. what is $N_P$?**
>
> We thank the reviewer for catching this oversight and apologize for the confusion. While we explicitly define:
>
> $N_D = \sum_{m=1}^M |{D}_m|$
>
> as the total number of decode tokens, we omitted the definition of $N_P$, which is the total number of prompt tokens:
>
> $N_P = \sum_{m=1}^M |P_m|.$
>
> It is somewhat clear from context, but it definitely should be defined explicitly. We have updated the paper to define $N_P$ where it first appears.
>
> > **How would the method generalize to different tasks?**
>
> RAC does not rely on any property specific to math or coding tasks. It only assumes access to (i) a distribution over prompts $x \sim D_{\text{task}}$ and (ii) the model’s own autoregressive rollout $\pi_\theta(\cdot \mid x)$. In general, for any task, RAC constructs calibration activations by sampling $T$ tokens for the CoT:
> $$z_{0:T} \sim \pi_\theta(\cdot \mid x), \qquad x \sim D_{\text{task}}$$
> and collecting the corresponding layer-input activations
> $$\mathbf{x}_t^{(\ell-1)}, \quad \ell = 1,\dots,L, t = 0,\dots,T$$
>
> for use in the layer-wise reconstruction objective. We would be excited to see others extend our work to, e.g., visual reasoning or other tasks, but this is beyond the scope of the current work.
>
>
> > **Is there any study on how the pruning varies with the CoT quality. With long reasoning traces, many times we observe the model has a long approach then backtracks and then corrects itself etc to get the output. Did you observe any specific behaviors there?**
>
> Thank you for your interesting question. To see if using lower quality CoT hurts the pruning performance, we have performed an experiment where we use the reasoning traces from Qwen3 1.7B model to prune the Qwen3 14B model. We compare this with pruning using the Qwen3 14B model traces below. We see that using the potentially lower quality traces from the 1.7B model indeed leads to suboptimal pruning quality, though the difference is modest.
>
> ### Pruning Qwen3 14B model with SparseGPT and RAC, using traces either from Qwen3 1.7B or Qwen3 14B. We report MATH500 accuracy.
> | Sparsity |  1.7B CoT | 14B CoT |
> | ------------------------------- |------------ | ------------ |
> | 30% |  0.960 (0.009) | **0.970 (0.008)**  |
> | 40% | 0.952 (0.01)  | **0.970 (0.008)**  |
> | 50% |  0.960 (0.009) | **0.962 (0.009)** |
>
> We thank the reviewer again for their helpful comments and hope our response and new results have addressed the comment you raised.

---

### Author Response · Authors · 2025-11-20
**Summary of New Experiments and Revisions**

We would like to thank all reviewers for their thoughtful comments that have helped us to improve the quality of our work. During the rebuttal period, we have performed several new experiments to address the comments and questions from the reviews. The major results from our experiments show:

- Robustness to model family: We perform extensive experiments with models from Qwen3 and Llama families, demonstrating that our proposed RAC benefits the pruning quality for these model families as well. These results are given in Tables 2, 3, and 4 of the revised manuscript.

- Robustness to pruning algorithm: We performed experiments with the ALPS [1] and WANDA [2] pruning methods. We observe that RAC also improves the pruning quality with these methods, in addition to SparseGPT. These results are given in Table 9 of the revised manuscript.

- New benchmark: We added AIME25 benchmark results for the Qwen3 family, in Table 4 (for SparseGPT) and Table 10 (for ALPS) in the updated manuscript. We see RAC also leads to better quality when we evaluate the pruned models on the AIME25 dataset, preserving accuracy very well even on this difficult mathematics benchmark.

We have highlighted the major changes to the paper in blue.

We also provided a point-by-point response to reviewers’ comments below, with some additional experiments. We thank the reviewers once again, and we look forward to hearing from them.

[1] ALPS: Improved Optimization for Highly Sparse One-Shot Pruning for Large Language Models, Xiang Meng, Kayhan Behdin, Haoyue Wang, Rahul Mazumder

[2] A Simple and Effective Pruning Approach for Large Language Models, Mingjie Sun, Zhuang Liu, Anna Bair, J.Zico Kolter

---

### Meta-Review · Area_Chair_Lqip · 2026-01-13

**Summary:**

This paper proposes Reasoning-Aware Compression (RAC), a simple but effective method for pruning reasoning LLMs. The key insight is that standard pruning calibrates on prompts or generic text (C4), but reasoning models spend most inference time on self-generated chain-of-thought tokens. RAC calibrates using the model's own on-policy CoT traces, aligning pruning with inference-time behavior. Strengths include: (1) simple, drop-in fix compatible with existing methods (SparseGPT, Wanda, ALPS); (2) strong results maintaining ~95% accuracy at 50% sparsity; (3) addresses a real problem where naively pruned reasoning models can become slower due to degenerate repetitive outputs. Weaknesses raised included limited initial experiments (only DeepSeek-R1), incomplete method comparisons, and concerns the contribution is too simple ("use task data for calibration"), though reviewers acknowledged simplicity can be a strength.

**Reviewer Concerns:**

Addressed: (1) Generalizability was comprehensively addressed with new experiments on Qwen3 (1.7B, 8B, 14B) and Llama (8B, 70B), showing RAC benefits are not specific to DeepSeek-R1. (2) Method comparison was completed with C4/prompt/RAC results for ALPS and Wanda in addition to SparseGPT, demonstrating algorithm-agnostic gains. (3) Harder benchmarks (AIME-25) were added, showing RAC maintains advantage on competition-level math. (4) Wall-clock overhead was quantified (~9 hours for trace collection, 0.25 hours for pruning on single GPU). (5) Difference from self-calibration [Williams et al.] was clarified: RAC uses task-conditioned CoT, not free-form BOS generation. (6) CoT quality experiment showed on-policy traces outperform traces from smaller models. Outstanding: The concern that the method is "too simple" remains, but simplicity enabling strong results is arguably a strength rather than weakness.

**Reviewer Scores:**

- bqFF (6 -> 6): Broadly positive reviewer whose concerns about notation and CoT quality were adequately addressed. Likely maintains score.
- hbbN (4 -> 4): Despite thorough clarification that RAC differs from "use task data" (prompt-only baseline is weaker), this reviewer admitted unfamiliarity with pruning literature and may remain unconvinced about contribution significance. Likely maintains score.
- LocM (4 -> 6): Explicitly stated willingness to increase score if generalizability experiments were provided. Rebuttal delivered exactly what was requested: Qwen3/Llama results, complete C4/prompt/RAC comparisons for ALPS and Wanda, and wall-clock overhead quantification. Would likely increase to 6.
- 1Hgg (6 -> 6): Concerns about self-calibration similarity, AIME benchmarks, and efficient reasoning comparisons were all addressed with detailed explanations and new experiments. Likely maintains positive score.

---

### Decision · Program_Chairs · 2026-01-26

Accept (Poster)